# Family and Community Obligations Motivate People to Immigrate—A Case Study from the Republic of the Marshall Islands

**DOI:** 10.3390/ijerph20085448

**Published:** 2023-04-10

**Authors:** Ryo Fujikura, Mikiyasu Nakayama, Daisuke Sasaki, Irene Taafaki, Jichao Chen

**Affiliations:** 1Faculty of Sustainability Studies, Hosei University, Chiyodaku, Tokyo 102-8160, Japan; 2Global Infrastructure Research Foundation, Minatoku, Tokyo 105-0001, Japan; 3International Research Institute of Disaster Science, Tohoku University, Sendai 980-8577, Japan; 4College of the Marshall Islands, Delap-Uliga-Djarrit, Majuro Atoll 96960, Marshall Islands; 5Graduate School of Public Policy and Social Governance, Hosei University, Chiyodaku, Tokyo 102-8160, Japan

**Keywords:** Marshall Islands, immigration, motivation, family and community obligations

## Abstract

A questionnaire survey was conducted in the Marshall Islands among 308 citizens of Majuro in order to analyze the factors that led them to immigrate. Using the results from the questionnaire items that indicate the motivations for emigration as independent variables, we extracted the factors with significantly high correlation coefficients; they suggest that the desire to escape from the many obligations within the family and regional community are predominant push factors for migrating overseas while the economic disparity between the United State and their home countries are predominant pull factors. Independently, the Permutation Feature Importance was used to extract the salient factors motivating migration, which provides similar results. Furthermore, the result of structural equation modeling verified the hypothesis that an escape from many obligations and economic disparity is a major motivation for migration at a significance level of 0.1%.

## 1. Introduction

### 1.1. Climate Change and Emigration

Environmental change is expected to affect both current and future migration through its impact on a wide range of economic, social, and political factors. The Intergovernmental Panel on Climate Change (IPCC) estimates that extreme weather events and other factors are already creating more than 20 million migrants worldwide today [1]. In the East Africa region, climate change could create up to 12.1 million migrants by 2050 [2]. While many patterns of migration caused by environmental change have been analyzed, the factors that determine migration are complex and it is very difficult to identify migration as primarily environmental [3,4,5,6,7]. While it is possible to identify migration that is primarily motivated by social and economic factors rather than the environment, there are cases where migration is accelerated by environmental change. When the United Nations Convention to Combat Desertification (UNCCD) conducted a survey in Morocco in 2017, most people were motivated to migrate due to a lack of employment opportunities; however, environmental change also played a major or partial role in their decision to migrate [8].

Small island states are vulnerable to climate change. At the IPCC at COP26, Tuvalu’s Foreign Minister made a video presentation from his country, knee-deep in seawater, about the threat of his country being submerged [9]. There is general concern that residents of small island states will be forced to leave their countries due to sea level rise. However, as discussed below, it has been reported that better education and employment are the main reasons for emigration from small island states, not necessarily environmental impacts.

For more than a decade, two of the authors have conducted research and studies on the resettlement [10] of residents due to dam construction and on the lives of those evacuated after the Fukushima Daiichi nuclear power plant accident in Japan [11]. They found that, contrary to popular perception, there were cases in which forced resettlement and evacuation led to improved lives after resettlement. In one case, the resettlement triggered by the construction of a dam enabled evacuees to escape their previous poor living conditions and harsh working environment, and they were highly satisfied with their lives after the resettlement [12]. Some of the evacuees from the Fukushima nuclear accident were able to escape the obligations of living in extended families by living in temporary housing, and they did not want to return even if their original housing was restored. Based on these experiences, we decided to conduct a questionnaire survey among residents of the Republic of the Marshall Islands (RMI), a small island nation, to determine whether there are other factors besides employment, education, medical care, and the environmental impact that motivate migration.

The following sections of this paper present the literature on Micronesian migration, the methodology of the survey and analysis, the results, and finally a discussion of the implications of the findings.

### 1.2. Micronesian Migration

Since the settlement of the Pacific islands more than 3500 years ago, islanders have migrated, expanded their settlements, and traded with neighboring tribes and nations [13]. As the rise in sea level becomes a serious threat to small island nations, migration is expected to increase in the future. Several studies have examined islanders’ motivations for migration. Neef et al. examined how indigenous Fijians’ adaptation strategies are influenced by values, resources, and information. They found that residents’ attitudes to climate-related risks vary, even within a small area that shares the same culture [14]. McMichael & Katonivualiku concluded that the relationship between climate impacts and displacement is uncertain but growing [15]. In the Republic of Nauru, an island country in the Central Pacific, the most likely reason for future migration is education, followed by family reunification. Planned relocation to adapt to climate change is expected to account for a significant proportion of future migration [16]. Fiji and Kiribati have begun to consider and implement “managed retreatment” to cope with the rise in sea level [17,18].

Kiribati is an atoll-only nation, and sea level rise is projected to make most of its land uninhabitable. Various programs are being implemented at the national level to mitigate the effects of climate change on livelihoods. Many households believe that they will have to migrate out of the country if the sea level rise—as well as the resulting saltwater intrusion, agricultural yields, and flooding—worsens. In particular, Kiribati people with university degrees tend to want to migrate abroad for further education and high-quality jobs. Kiribati migrants to Fiji cited climate change as a factor in their migration, although the main reason was to seek better education and employment. In coastal areas facing long-term risks from sea level rise, the relocation of housing and infrastructure is a viable policy option [19,20].

Tuvalu, like Kiribati, is an atoll nation; it was noted that climate change would likely lead to a large-scale migration from it. However, for most people in the country, climate change was not a cause for concern, let alone a reason to migrate. Potential migrants also did not cite climate change as a reason to migrate. People in the capital—Funafuti—want to stay for lifestyle, culture, and identity reasons [21].

Residents of the Carteret Islands in Papua New Guinea are being relocated to the nearby island of Bougainville after it was determined that relocation off the island was inevitable due to land subsidence. However, conflicts with the host society have arisen at the relocation site, leading some people to return to their original island [22].

## 2. Materials and Method

### 2.1. Marshall Islands

The Republic of the Marshall Islands (RMI) is an island nation in Micronesia that consists entirely of atolls. The population grew up until 2004, reaching 54,435, but has since declined to 42,050 in 2021 [23]. The highest elevation is only two meters above sea level. The effects of climate change are already being felt. The RMI is clearly more vulnerable to climate change than the non-atoll countries. Rising sea levels threaten to make much of the country uninhabitable. The coastline of Majuro, the capital, is frequently inundated by waves from both the North and South Pacific. At times, storm surges can completely inundate the island [24]. Since 2008, approximately 2046 people have been evacuated due to storm surges [25].

The RMI, along with the Federated States of Micronesia and the Republic of Palau, has a Compact of Free Association (COFA) with the United States, which allows citizens of the three member countries to live and work there without a visa. The RMI’s COFA expires in 2023, but negotiations are currently underway to extend it [26]. The number of Marshallese living in the United States has grown from 6650 in 2000 to an estimated 30,000 in 2018. They are concentrated in three states: Hawaii, Arkansas, and Washington [27].

While Marshallese have come to the United States in search of education, family integration, jobs, and health care, many of them are concerned about climate change. Drinkall et al. surveyed Micronesians living in Oregon and found that 95% believe that climate change will have a negative impact on their home country [28]. A survey by van der Geest et al. found that many Marshallese in Hawaii indicated that environmental issues motivated their migration [24] and 43% of Marshallese living in the United States indicated that environmental factors prompted their decision to migrate [29].

Oakes et al. suggest that environmental factors are already a factor in the decision to relocate to the United States, even for Marshallese living in the RMI [19]. However, while some studies acknowledge that environmental change is stressful for residents, it is not a significant motivator for migration; education, employment, family, and healthcare remain important motivators [30,31,32].

Possible climate change adaptation measures that the RMI could take include migration to developed countries, as well as migration to neighboring countries or to artificial islands built by raising the land, as seen in the Maldives [33]. As long as COFA is in effect, migration to the United States is considered a better option [34]. This study aims to determine what motivates their migration by conducting a questionnaire survey of Marshallese living in Majuro, the capital of the RMI. In addition to education, employment, and medical care—which were previously identified as important factors—this study added local religious and social factors among the ones influencing their migration motives. The predominant religion in the RMI is Protestantism,—suggesting that these beliefs, traditional culture, and spirituality influence feelings about the rise in sea level and homeland [35,36,37]. In addition, the Micronesian region—including the RMI—has a tradition of living in extended families and tends to recognize and demand that people fulfill their obligations to them [38,39,40]. Because the motivation to escape social pressures in one’s home country may be a possible motivation for migrating to another country, related questions were added to the survey items.

The results of the questionnaire survey reveal that the desire to escape from the many obligations within the family and community, which had not previously been considered as a cause of migration, was a strong motivator for Majuro citizens to migrate abroad. This paper presents the methodology and results of this questionnaire survey, as well as the possible policy aspects that can be considered.

### 2.2. Questionnaire Survey

A workshop with Marshall Islands-based researchers, International Organization for Migration (IOM) staff, and researchers from the United States was held with this study’s authors in Tokyo on 23 January 2020 to finalize the English version of the questionnaire (Appendix A).

The questionnaire asked about respondents’ demographics (age, gender, marital status, education, household income, and occupation), followed by their perceptions of climate change, trust in information sources, and relationship to religion (Q1–Q4, Q7–Q15). Q16 indicates the strength of the intention to emigrate overseas. Q19 is the perception of environmental issues other than climate change. Except for Q19, all questions from Q17 onward are items that could be related to motives for overseas emigration. That is, English proficiency (Q17), ability to work abroad (Q18), opportunities to discuss emigration (Q20, Q21), number of (extended) family members living abroad (Q22), decisions about land ownership and use (Q23), obligations within the family and community (Q24, Q5), medical needs (Q26), economic motivations (Q27), confidence in ability to maintain their identity as Marshallese abroad (Q28), sending money home (Q29), possible career choices (Q30), the tendency of people who went abroad to stay there (Q31), and expectations that emigration will improve their lives (Q32).

The questionnaire was translated into Marshallese and a random sample of Majuro residents were interviewed by Marshallese researchers. The sample size was set at 300 to ensure a sampling error of 5% and a confidence interval of 92%, resulting in a sample size of 308. The survey period was from July to December 2021. The main demographic characteristics of the subjects are presented in Appendix B.

The results of the tabulation were used to determine the correlation coefficients between Q16, which indicates the strength of the intention to emigrate overseas, and other question items.

### 2.3. Identification of Dominant Factors by Permutation Feature Importance

We attempted to identify the dominant explanatory variables using the Permutation Feature Importance (PFI). The PFI measures the extent of increase in the prediction error of the model after rearranging the feature values and breaking the relationship between the feature and the true outcome [41].

The importance of a feature is measured by calculating the amount of increase in the model’s prediction error after rearranging the feature values. A feature is “important” if shuffling its value increases the model’s error because the model relies on that trait to make predictions. The model makes predictions without considering the feature, so if the shuffling does not change the model error, the feature is considered “unimportant”.

### 2.4. Structural Equation Modeling (SEM)

Structural Equation Modeling (SEM) can represent the entire structure of the relationship between variables and be used to assess “latent variables” that are not observable, such as one’s subconscious. According to Tarka (2018), we can measure latent variables indirectly, mostly by using a set of observable variables and by observing the causal effects in SEM between the respective latent variables [42]. SEM can also be used in a confirmatory factor analysis (CFA) to test whether the hypothesis is true or not. A study by Sasaki et al. had previously applied SEM to a questionnaire survey conducted in Majuro [43]. In this study, the authors use SEM to test the hypothesis that dissatisfaction with the many obligations within the family and community is a strong motivator for Majuro citizens to migrate abroad. The software package used for SEM in this study was SPSS Amos 28. 

## 3. Results

### 3.1. Questionnaire

Out of the total respondents, 75.5% reported having one or more family members living abroad (Table 1). Given the large family system in the Marshall Islands, this suggests that many family members are living abroad. Question 16 (“I wish to migrate abroad sometime in the future.”) asked respondents about their intention to migrate abroad, in which 45.1% of respondents said they did and 40.5% said they did not—an almost even split (Table 2).

### 3.2. Factors Determining Migration

Table 3 lists question items that correlate with question 16 at a significance level of 99% or greater.

As in previous studies, economic reasons (Q27, Q29, Q32), family (Q22), and medical care (Q26) are shown to be motivating factors for migration; however, the magnitude of the correlation coefficients suggests that family (Q24) and community (Q25) obligations are also identified as strong motivators.

Church attendance (Q11) was weakly correlated with motivation to migrate, but no significant relationship was found with the other questions indicating strength of religious belief (Q12, Q13, Q14, and Q15). No significant relationships were found for any of the questions indicating perceptions or concerns about climate change, including sea level rise (Q1, Q2, Q3, Q4, Q7, Q8, Q9, and Q10). This contrasts with Marshallese living in the United States, who cited climate change as a reason for coming to the United States or not returning [29,30].

### 3.3. Contribution of Independent Variables Estimated by Permutation Feature Importance

The above analysis identified three items with correlation coefficients above 0.4 with question 16, which asks about economic motivations for migrating abroad; this suggests that strong family and community obligations and household financial instability are strong motivators. To further validate these results, the following two steps are used to identify the “outstanding factors” that determine migration.
(1)A model is constructed using the XGBoost (eXtreme Gradient Boosting) algorithm being implemented in the R’s “caret” package, where the answer to question 16 is the dependent variable and the answers to all other questions are the independent variables.(2)Permutation Feature Importance (PFI) is applied to the calibrated model to identify the independent variables that are the “outstanding factors”.

To build the model, the answers to all questions except Q16 (which is the dependent variable) are used as independent variables. Three question items (Q11, Q22, and Q23) whose responses are not based on a five-point Likert scale method are excluded, so the number of independent variables is 28.

In PFI, the independent variable importance is the reduction in the model’s estimation accuracy that occurs when the values of an independent variable are completely reordered using random numbers. If the reordering causes a significant change in the model’s performance (i.e., if the model’s error rate increases substantially), then the independent variable is considered important. Conversely, if the reordering has little effect on the model’s output (i.e., the model’s error rate does not change much), then the independent variable is considered unimportant. 

Since the dependent variables are discrete values (from one to five) using the Likert scaling method, the estimation error is 0% when the output of the identified model is rounded to an integer.

As a result, the five items shown in Table 4 are the top five “outstanding factors” with the highest level of significance (i.e., the magnitude of the error rate of the model). These are all questions with correlation coefficients above the 99% significance level shown in Table 3. Again, this suggests that the strength of family and community obligations—as well as economic reasons—motivate relocation.

### 3.4. Verification of Hypotheses by Structural Equation Modeling

The previous evidence implies that economic reasons and strong family and community obligations may have a significant impact on migration intentions. Here, we confirm if this hypothesis is correct by using structural equation modeling (SEM). A study by Sasaki et al. had previously applied SEM to a questionnaire survey conducted in Majuro [43].

The analysis is based on the five independent variables (Q24, Q25, Q27, Q29, and Q32) and the dependent variable (Q16), all of which were rated highly important by the PFI. Based on the above hypotheses, we constructed a path diagram and obtained the results shown in Figure 1. The answers to Q24 and Q25 might indicate the degree of dissatisfaction with the status quo, while Q16, Q27, Q29, and Q32 could be interpreted as questions about the desire to migrate abroad.

All path coefficients were significant at the 0.1% level, and the standardized coefficient for the latent variable (from dissatisfaction to international migration desire) was 0.86, indicating that it has a substantial impact. Thus, the authors conclude that the result of the SEM verified the hypothesis that a dissatisfaction with family and community obligations and economic motives influence the intention to migrate abroad.

As for the goodness of fit index, the Root Mean Square Error of Approximation (RMSEA) was slightly poor at 0.061; however, the results for the other indices were generally satisfactory.

## 4. Discussion

Many studies have shown that there are many factors that determine relocation and which are highly interrelated. Lee (1966) proposed a theory of migration by presenting the factors that promote relocation and suggesting that it consists of push factors that push people out of their place of origin and pull factors that pull them to their destination [44]. This is widely accepted as the push-and-pull factor in migration [45]. Destination countries attract people through the disparity between the economies of origin and destination [46,47] and the associated remittances to the home country [48,49]. These can be pull factors for migration. On the other hand, hardship or dissatisfaction at home caused by conflict, environmental degradation and poverty make it easier for people to leave. These can be push factors.

In the case of the RMI, 45% of Majuro residents indicated the possibility of moving abroad; this study showed that the dominant push factors were the motivation to break away from strong family and community obligations (Q24, Q25). In Micronesian countries, including the RMI, people live in extended families. It is possible that the people in the region, like the citizens of Majuro, feel a strong sense of obligation because of the extended family system and move out of the country because they wish to distance themselves from it. In addition, economic inequality was seen as the dominant pull factor (Q27, Q29, Q32). Environmental impacts, including sea level rise, could be a pull factor, but that did not seem to be the case here. 

Also, although not included in the PFI as an important outcome, the desire to receive medical care in a developed country should also be noted. The Marshall Islands face a number of health challenges, including high rates of obesity and cancer among non-communicable diseases, lack of specialized medical services, and the third-highest prevalence of diabetes in the world [50]. This fact makes it natural that they would want to live in the United States, whether for the short or long term.

## 5. Conclusions

Since there is no foreign country in the Maldives to which the people could emigrate unconditionally, they have no choice but to remain in their own country. The Maldivian government’s policy of relocating its citizens to newly developed land in the country reflects this situation. The land development project cost $400 million and was financed primarily by loans from Saudi Arabia, China, and the United Arab Emirates (UAE). As a result, the Maldives has incurred significant debt—which has caused both local and international concern [51]. 

In contrast, the RMI citizens are free to emigrate to the United States as long as the COFA is in effect. Therefore, it is likely that they will continue to migrate to the United States because they dislike their obligations resulting from the extended family system. Unfortunately, the lives of Marshallese living in the United States are often socially and economically disadvantaged [52]. One reason for this is a lack of preparation for emigration. Many Marshallese who readily accept invitations to immigrate from relatives living in the United States come to the country without adequate English language skills or job training. They have difficulty adjusting to the host society after immigration, and are often compelled to continue the extended family system in the destination country. The presence of an extended family in the destination country means that a foundation for life is already in place immediately after migration; this has the effect of solving or alleviating many of the problems that may arise as a result of migration. On the other hand, immigrants will not be able to escape the obligations that come with the extended family system, even after migration. In addition, the inability to integrate into the local community and the discrimination they face will cause them more stress than they feel in their home countries.

From the perspective of the RMI national management, COFA also encourages the outflow of skilled and talented people from the RMI to the United States [49]. If they acquire skills in the United States and then return home, this may facilitate the development of the RMI. However, as noted above, Marshallese in the United States have little intention of returning home due to concerns about sea level rise and other climate changes [27,28,29]. This results in increased vulnerability as a state of the RMI. So far, there appears to be little discussion within the RMI regarding migration policy in light of these negative effects of COFA.

On the other hand, 41.5% of respondents in this study had a negative view of emigration (Table 2). In Pacific countries, people’s lives are integrated with the land and they have very strong ties to it [53]. They want to stay in the RMI in the future. Oberman questions the use of migration as a means of alleviating global poverty [54]. He argues that the poor have a human right to remain in their own countries and receive assistance without having to migrate abroad. Cubie also points out that not all people affected by climate change have the will or ability to leave high-risk areas [55]. Policymakers should respect the rights of those who wish to remain when implementing adaptation measures, Cubie suggests, and non-migration must be considered as a strategy for coping with climate change. 

While the purpose of this study was to explore the motivations of those who wanted to emigrate, it was not possible to explore those who did not want to emigrate, which represented 41.5% of the respondents. The risk that a significant part of the country will become uninhabitable in the future is considered quite high. What they would like to do in such a situation is the next question to be explored.

One option for protecting existing islands is to build seawalls to reduce the effects of erosion and storm surges. The cost in the RMI is estimated at USD $100 million, which is nearly twice the annual national income of the RMI [51]. Financial support from the international community is essential. Even so, the problem of the obligations of the extended family system would not be eliminated. Even if the RMI can elevate the land (as in the Maldives), and the Marshallese are able to settle on the newly created land, it is likely that the extended family system and community customs will remain intact. Whether they would be able to live satisfactorily is uncertain.

The Sharm el-Sheikh Implementation Plan [56], which was adopted at the UNFCCC at COP 27, included migration as a consideration in loss and damage. However, the factors driving migration are many and complex. It is quite possible that investing international funds in a project to reduce the incidence of migration without fully studying the social situation may not be as effective as hoped, because it is not known whether people are actually moving for other reasons. 

Discussions of loss and damage from climate change would only consider climate migrants who are involuntarily displaced for compensation, while excluding voluntary migrants from compensation. In reality, however, climate change is not necessarily the dominant factor in the decision to migrate, and in many cases other factors dominate. In such a situation, if only those people who are likely to migrate in the future and for whom climate change is the predominant cause of migration are eligible for compensation, social injustice could result. Even among people who do migrate, there could be inequities; some may receive compensation and others may not, depending on the timing of their migration. It is possible that the emergence of a large number of people who will not migrate until they can receive compensation could become a social problem.

It is important to fully assess whether the people affected by climate change really want to stay where they are or whether they are, in fact, willing to migrate even if they are not affected by climate change. On the other hand, countries and regions that receive migrants must carefully accommodate them to support “migration with dignity,” so that they can successfully rebuild their lives in their new location [57]. There is no one-size-fits-all policy. It is necessary to present as options policies that are best suited to different realities.

## Figures and Tables

**Figure 1 ijerph-20-05448-f001:**
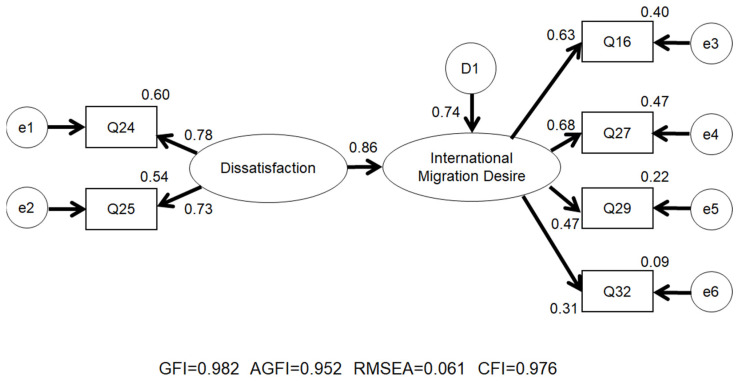
Path diagram as a result of the SEM.

**Table 1 ijerph-20-05448-t001:** Family members living abroad.

0	73
1–5	46
6–10	31
11–15	20
16–20	128
20+	0
Total	298

**Table 2 ijerph-20-05448-t002:** I wish to migrate abroad sometime in the future.

Strongly disagree	56
Disagree	71
Neither agree nor disagree	41
Agree	85
Strongly agree	53
Total	306

**Table 3 ijerph-20-05448-t003:** Questions correlated at a significance level of 99% or higher with motivation to migrate.

No.	Question	Correlation Coefficient
Q24	I may migrate because I have too many obligations as a member of my family.	0.417
Q27	I may migrate because of financial insecurity.	0.405
Q25	I may migrate because I have too many obligations as a member of the community I belong to.	0.400
Q29	The financial well-being of my family is maintained by the money sent by a family member living abroad.	0.343
Q11	How many times a week do you go to church?	0.240
Q32	People who move abroad have a better life.	0.238
Q26	I may migrate because I or a family member might require medical attention.	0.217
Q22	How many of your family members live in abroad?	0.152
Q19	Serious environmental issues (such as pollution, waste management, and dengue) exists other than climate change in RMI.	−0.195

**Table 4 ijerph-20-05448-t004:** Questions with large error rates.

Question	Increase of Error Rates (%)	Correlation Coefficient (Shown in Table 3)
Q24	30.9	0.417
Q29	30.0	0.343
Q27	26.6	0.415
Q25	24.5	0.400
Q32	14.5	0.238

## Data Availability

The data are available from the corresponding author upon reasonable request.

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
