# Peer review of "Family and Community Obligations Motivate People to Immigrate—A Case Study from the Republic of the Marshall Islands"

_ijerph, 2023, doi:10.3390/ijerph20085448_

Round 1
Reviewer 1 Report
The manuscript explores the factors that lead Marshall Islands citizens to immigrate. The topic si interesting but the paper's structure should be substantially revised as explained below.
Introduction should be completely re-orgnised. After presenting the relevance of the topic and discussing the research motivation, the Authors should stress the gap in the literature that they intend to fill and clearly present the research question(s) that they aim to answer. At the moment these elements are missing. Then they should shortly (in a few lines) describe the case study they investigate and the methodology employed. Finally the section should end with the paper’s remainder in order to guide the reader throughout the manuscript.
Section 1.2 should be moved into a sub-section of section 2 that, in turn could be renamed as “materials and methods”. In this section, the Authors can provide information about the context of analysis (Marshall Islands) and then describe the methodology.
Since a literature section is missing in the manuscript, I suggest to reinforce the literature provided in introduction.
The methodological section is poor. The Authors have limited to explain that they have collected data by administering a questionnaire. Although this is reported in full in Appendix, I believe that a short discussion here about the questions included in the questionnaire can be useful for the reader. Later on in the manuscript (section 3), the Authors reveal that they have implemented Permutation Feature Importance. I believe that this information should be moved into the section 2, leaving section 3 only for presenting the findings. In contrast, Tables 1 and 2 represent already results and, accordingly, they should be moved into section 3. Summing up: any information provided in section 3 (and subsections 3.x) concerning the methodology should be moved to section 2.
Conclusions should further stress the policy implications of the study, the limitations and the replicability of the results.
Suggested references:
(2008) Journal of Urban Economics, 64 (3), pp. 519-537
(2015) World Development, 71, pp. 94-106
(2023) Regional Studies, Regional Science, 10(1), pp. 1–19
Author Response
Thank you very much for your constructive suggestions. Please find attached a revised version.
Introduction should be completely re-orgnised. After presenting the relevance of the topic and discussing the research motivation, the Authors should stress the gap in the literature that they intend to fill and clearly present the research question(s) that they aim to answer.
I have added why we conducted this study. It is based on our previous studies of dam resettlement and nuclear power plant accident evacuees.
Then they should shortly (in a few lines) describe the case study they investigate and the methodology employed. Finally the section should end with the paper’s remainder in order to guide the reader throughout the manuscript.
At the end of the first section of the introductory chapter (lines 79-86), I described.
Section 1.2 should be moved into a sub-section of section 2 that, in turn could be renamed as “materials and methods”. In this section, the Authors can provide information about the context of analysis (Marshall Islands) and then describe the methodology.
I have rewritten it as you suggested.
Since a literature section is missing in the manuscript, I suggest to reinforce the literature provided in introduction.
I have summarized the literature in Section 1.2 and Section 2.2.
I believe that this information should be moved into the section 2, leaving section 3 only for presenting the findings. In contrast, Tables 1 and 2 represent already results and, accordingly, they should be moved into section 3. Summing up: any information provided in section 3 (and subsections 3.x) concerning the methodology should be moved to section 2.
I have rewritten as you suggested. I improved the methodology sections.
Conclusions should further stress the policy implications of the study, the limitations and the replicability of the results.
I tried to improved the conclusion part. I added references that you suggested.
English will be proofread after acceptance.

Reviewer 2 Report
The paper entitled “Family and Community Obligations Motivate People to Immigrate. A Case Study from the Republic of the Marshall Islands”, aims to analyze the factors that lead Marshall Islands citizens to emigrate. This is an interesting study because the rise in sea level because of climate change is a serious threat to Micronesia. Accordingly, migration is expected to increase in the future in this region.
At the methodological level, from July to December 2021 308 randomly selected citizens living in Majuro City were surveyed. I want the authors to explain why they selected 308 people. I guess that the authors intended to reach a 5% sampling error and a 92% confidence interval (1.81 z score in a normal curve). Also, I would like to know more in-depth how the authors constructed the sample. The authors explain that they randomly selected the sample studied. However, they do not explain what probability sampling method was used in this study: simple random sampling, systematic sampling, stratified sampling, or cluster sampling.
On the other hand, the theoretical underpinning of the paper is weak. This study aims to analyze the factors that lead Majuro City islanders to emigrate. There is a wide corpus of theories explaining why people emigrate. The authors conclude that the desire to escape from the many obligations within the family and regional community is a predominant factor for migrating overseas. Also, as it is pointed out in Q29, the financial well-being of the migrants’ families is maintained by the money sent by a family member living abroad, or as it is pointed out in Q26 interviewees would consider emigrating if a family member requires medical attention. Therefore, the new economics of labor migration theory could be very useful to explain Q29 or Q26. The most important (most cited) papers on the new economics of labor migration theory are: Stark, O., & Bloom, D. E. (1985). The new economics of labor migration. The American Economic review, 75(2), 173-178, and Taylor, E. J. (1999). The new economics of labour migration and the role of remittances in the migration process. International migration, 37(1), 63-88. Additionally, as it is pointed out in Q32, respondents think that people who move abroad have a better life, and as it is pointed out in Q27 interviewees are intending to migrate because of financial insecurity. Accordingly, Neoclassical migration theory can help to explain Q32 and Q27.
In conclusion, this is an interesting paper, however, the authors must explain more in-depth the methodology used and must correct the theoretical weakness of the article.
Author Response
Thank you very much for your constructive suggestions. Please find attached a revised version.
I want the authors to explain why they selected 308 people. I guess that the authors intended to reach a 5% sampling error and a 92% confidence interval (1.81 z score in a normal curve). Also, I would like to know more in-depth how the authors constructed the sample. The authors explain that they randomly selected the sample studied. However, they do not explain what probability sampling method was used in this study: simple random sampling, systematic sampling, stratified sampling, or cluster sampling.
The method used was simple random sampling. The sample size was set at 300 to ensure a sampling error of 5% and a confidence interval of 92%, resulting in a sample size of 308. This is mentioned in the questionnaire section (lines 254-256)
On the other hand, the theoretical underpinning of the paper is weak.
Referring to the literature you suggested, I used the push and pull factors model to explain the immigration mechanism in the discussion chapter (lines 607-617). While economic reasons are pull factors, obligations at home are push factors.
English will be proofread after acceptance.

Reviewer 3 Report
It is of great value that, in addition to other factors that have been previously identified as important, this study adds local religious and social factors among the ones influencing their migration motives. The methodology is adequate to confirm the hypothesis.
Regarding the methodology, it would be necessary to specify how the sampling was carried out.
Regarding the section of results, authors should consider changing the writing of lines 140-143 to adjust the analysis more to the results, since it is not mentioned that the correlation of Q27 is greater than the one of Q25.
Regarding the section of discussion, it would be necessary to mention the limits of the research and future lines, such as, for example, qualitative studies (interviews) that ensure that all the variables that generate interest towards migration in that area have been considered.
Regarding more specific changes, in Tables 1 and 2, the titles should be edited so that they have the same structure: remove Q16 in Table 2 and add a period at the end of the sentence in Table 1. In Table 3 there is also a sentence without a period.
Due to the wording in lines 227-232, it seems that statements are made without a citation. I assume that the source of those statements is [39], but a change in the wording would avoid misunderstandings.
I recommend changing ‘possible policy responses that can be considered’ to ‘possible policy aspects that should be considered’, as I consider that discussion and conclusions are more referred to comment policy aspects that must be considered than presenting possible policy responses that can be considered.
Author Response
Thank you very much for your constructive suggestions. Please find attached a revised version.
Regarding the methodology, it would be necessary to specify how the sampling was carried out.
The method used was simple random sampling. The sample size was set at 300 to ensure a sampling error of 5% and a confidence interval of 92%, resulting in a sample size of 308. This is mentioned in the questionnaire section (lines 254-256).
Regarding the section of results, authors should consider changing the writing of lines 140-143 to adjust the analysis more to the results, since it is not mentioned that the correlation of Q27 is greater than the one of Q25.
I rewrote the sentence as follows: however, the magnitude of the correlation coefficients suggests that family (Q24) and community (Q25) obligations are also identified as strong motivators.
Regarding the section of discussion, it would be necessary to mention the limits of the research and future lines, such as, for example, qualitative studies (interviews) that ensure that all the variables that generate interest towards migration in that area have been considered.
I added one paragraph mentioning future research in the Conclusion Chapter as below; “While the purpose of this study was to explore the motivations of those who wanted to immigrate, it was not possible to explore those who did not want to immigrate, which represented 41.5% of the respondents. The risk that a significant part of the country will become uninhabitable in the future is considered quite high. What they would like to do in such a situation is the next question to be explored."
Regarding more specific changes, in Tables 1 and 2, the titles should be edited so that they have the same structure: remove Q16 in Table 2 and add a period at the end of the sentence in Table 1. In Table 3 there is also a sentence without a period.
I rewrote.
Due to the wording in lines 227-232, it seems that statements are made without a citation. I assume that the source of those statements is [39], but a change in the wording would avoid misunderstandings.
I rewrote.
I recommend changing ‘possible policy responses that can be considered’ to ‘possible policy aspects that should be considered’, as I consider that discussion and conclusions are more referred to comment policy aspects that must be considered than presenting possible policy responses that can be considered.
I rewrote.
English will be proofread after acceptance.

Round 2
Reviewer 1 Report
Dear Authors, I found the manuscript definitely improved. In my opinion, it has reached an adequate level of scientific soundness and therefore it is ready for publication. Congratulations!